# CHANNEL-PRIORITIZED CONVOLUTIONAL NEURAL NETWORKS FOR SPARSITY AND MULTI-FIDELITY

**Chun-Min Chang**
National Taiwan University
d05921027@ntu.edu.tw

**Hung-Yi Ou Yang**
National Cheng Kung University
frank840925@gmail.com

**Chia-Ching Lin**
National Taiwan University
d05921018@ntu.edu.tw

**Chin-Laung Lei**
National Taiwan University
cllei@ntu.edu.tw

**Kuan-Ta Chen**
Academia Sinica
swc@iis.sinica.edu.tw

## ABSTRACT

We propose a novel convolutional neural networks (CNNs) training procedure to allow dynamically trade-offs between different resource and performance requirements. Our approach prioritizes the channels to enable structured sparsity and multi-fidelity approximations at inference time. We train the VGG network with our method on various benchmark datasets. The experiment results show that, on the CIFAR-10 dataset, a $63\times$ parameters reduction and a $11\times$ FLOPs reduction can be achieved, with only a 2% accuracy drop.

## 1 INTRODUCTION

While the trend nowadays is to make neural network architectures deeper to improve performance, it is still desirable to deploy a compact and computational efficient model with multi-fidelity approximations to allow dynamic scaling over a computation range, especially for applications on end devices. The term *fidelity level* here is defined as the proportion of features (channels in the case of CNNs) applied during forward propagation. For example, a 30% fidelity level approximation uses 30% of the channels in every layer to obtain an approximated result. We can use the entire set of features for inference under normal condition, and a lower fidelity level approximation under resource limited condition.

Many computational efficient networks focus on reducing the model size, either by directly forcing weights to become zeros in Han et al. (2016); Li et al. (2016); Molchanov et al. (2016), or by inducing zeros in the scaling factors in the batch normalization layers by Liu et al. (2017); Ye et al. (2018). Other techniques, such as low-rank approximation by Denton et al. (2014); Jaderberg et al. (2014), weight grouping by Gong et al. (2014); Han et al. (2015); Zhou et al. (2017); Ullrich et al. (2017), and group sparsity regularizer by Wen et al. (2016); Zhou et al. (2016); Alvarez & Salzmann (2016) can be used independently or in conjunction with other approaches to further reduce the model size. However, none of these techniques supports dynamic scaling to meet varying resource and performance requirements. On the other hand, as for adaptive computation networks, the work by McDanel et al. (2017) introduces a global scaling factor to dynamically control the percentage of features being included. Huang et al. (2017) proposes a network architecture framework that uses a cascade of intermediate classifiers throughout the network with early-exit ability.

Our approach, *channel prioritization*, is a two-stage procedure that benefits from both design considerations above. We prioritize the channels in each layer by their channel indices. Before training, all scaling factors of each batch normalization (BN) layer are initialized with a monotonically decreasing function. In the training stage, we incorporate L1 penalty and monotonicity-induced penalty on the scaling factors to encourage sparsity and keep the monotonicity, as will be discussed in Section 2. After training, we prune away insignificant channels and then fine-tune the network again. In the fine-tuning stage, we also aggregate the losses at different fidelity levels to achieve multi-fidelity approximations.

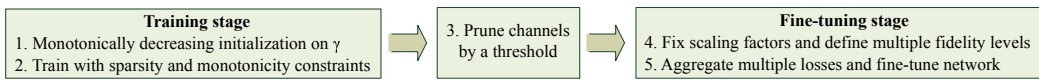

**Figure 1:** Flow-chart of channel prioritization procedure.

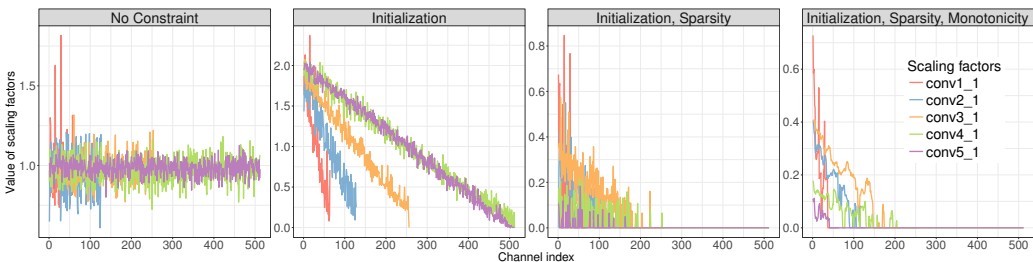

**Figure 2:** The scaling factors trained under different constraints on the CIFAR-10 dataset. From left to right, the percentage of the values smaller than 0.05 is 0%, 1.9%, 78%, 77%, and the Kendall's rank correlation is 0.02, -0.86, -0.39, and -0.66.

## 2   CHANNEL PRIORITIZATION

We leverage the fact, that channel importance becomes comparable across the network since every scaling factor in batch normalization (BN) layers Ioffe & Szegedy (2015) always multiples a normalized activation, to design the following initialization and penalty techniques that prioritize channels. Channel prioritization enables to prune networks in a structured way without sorting, and provides a simple and effective way to control the percentage of parameters and resource involved.

**(C1) Monotonically Decreasing Initialization.** We initialize the scaling factors with a monotonically decreasing function as follow to induce priority along the channels. Given the $l$-th layer with $N_l$ channels,

$$\gamma_l^{(k)} = 2 \times (1 - \frac{k-1}{N_l}), \quad k = 1, ..., N_l \tag{1}$$

**(C2) Sparsity penalty.** We impose L1 penalty on scaling factors, $|\gamma_l^{(k)}|$ to induce sparsity.

**(C3) Monotonicity-induced penalty.** We quantify monotonicity as the successive difference between any two consecutive scaling factors. The penalty is defined as follow.

$$L_{m,l}^{(k)} = \begin{cases} \gamma_l^{(k+1)} - \gamma_l^{(k)} & \text{, if } \gamma_l^{(k+1)} > \gamma_l^{(k)} \\ 0 & \text{, otherwise} \end{cases}, \quad k = 1, .., N_l - 1 \tag{2}$$

In the training stage, the objective is the weighted sum of cross-entropy loss, sparsity penalty, and the monotonicity-induced penalty on scaling factors is shown below.

$$L_{obj} = Loss + \lambda_s \sum_{k,l} |\gamma_l^{(k)}| + \lambda_m \sum_{k,l} L_{m,l}^{(k)} \tag{3}$$

where $\lambda_s$ and $\lambda_m$ tradeoff between loss, sparsity and monotonicity. In this practice, we aim to sparsify and prioritize the scaling factors, which allows pruning insignificant channels afterwards with ease.

**Multi-fidelity Approximations.** Prioritized channels enable us to select the foremost $p\%$ in the channel index of every layer to compute the $p\%$ fidelity approximation. We aggregate the losses at various fidelity levels as the fine-tuning objective to optimize multi-fidelity approximations. Given a set of fidelity levels, $P$

$$L_{obj} = \sum_p Loss_p, \forall p \in P \tag{4}$$

, where and the loss at $p\%$ fidelity level is denoted as $Loss_p$. In the fine-tuning stage, we fix the parameters in the BN layers and update the others. Thus, there is no constraints on sparsity and monotonicity. The complete algorithm is summarized as Figure 1.

**Table 1:** Performance and resource demands

| Dataset | Model | Accuracy (%) | Parameters | Ratio (%) | FLOPs | Ratio (%) |
|---------|-------|--------------|------------|-----------|-------|-----------|
| CIFAR-10 | VGG-16, baseline | 88.7 | $1.50 \times 10^7$ | - | $6.26 \times 10^8$ | - |
| | Li et al. (2016) | $+0.2$ | $5.40 \times 10^6$ | 36.0 | $4.12 \times 10^8$ | 65.8 |
| | Liu et al. (2017) | $+0.2$ | $1.72 \times 10^6$ | 11.5 | $3.13 \times 10^8$ | 51.0 |
| | Our, IDP at 100% | $+0.5, -5.8$ | $9.10 \times 10^5, 1.50 \times 10^7$ | 6.1, 100 | $1.81 \times 10^8, 6.26 \times 10^8$ | 28.9, 100 |
| | Our, IDP at 75% | $-0.7, -6.1$ | $5.17 \times 10^5, 8.49 \times 10^6$ | 3.5, 56.6 | $1.08 \times 10^8, 3.68 \times 10^8$ | 17.4, 58.8 |
| | Our, IDP at 50% | $-2.7, -8.2$ | $2.40 \times 10^5, 3.83 \times 10^6$ | 1.6, 25.5 | $5.58 \times 10^7, 1.79 \times 10^8$ | 8.9, 28.6 |
| | Our, IDP at 25% | $-12.8, -79.2$ | $6.92 \times 10^4, 1.00 \times 10^6$ | 0.4, 6.67 | $2.00 \times 10^7, 5.70 \times 10^7$ | 3.2,, 9.10 |
| CIFAR-100 | VGG-16, baseline | 64.0 | $1.50 \times 10^7$ | - | $6.26 \times 10^8$ | - |
| | Liu et al. (2017) | $+0.3$ | $3.78 \times 10^6$ | 25.2 | $3.90 \times 10^8$ | 62.4 |
| | Our, IDP at 100% | $+0.5, -0.3$ | $2.99 \times 10^6, 1.50 \times 10^7$ | 19.9, 100 | $3.48 \times 10^8, 6.26 \times 10^8$ | 55.6, 100 |
| | Our, IDP at 50% | $-7.3, -30.0$ | $8.20 \times 10^5, 3.87 \times 10^6$ | 5.5, 25.8 | $1.08 \times 10^8, 1.79 \times 10^8$ | 17.3, 28.6 |

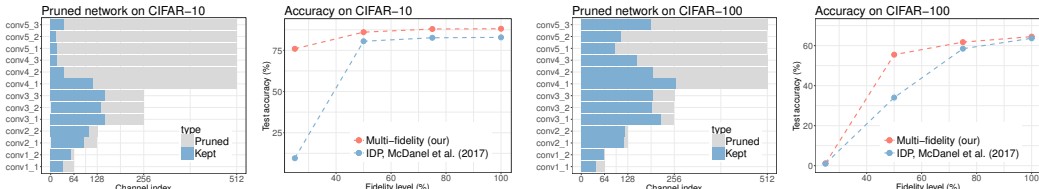

**Figure 3:** Visualization of the pruned network structures and fine-tuned performance on the CIFAR-10 (left) and CIFAR-100 (right).

# 3 EXPERIMENTAL RESULTS

We test our approach using the ImageNet pre-trained VGG-16 model on the CIFAR-10 and CIFAR-100 datasets. For the BN layers, we initialize the scaling factors by equation 2, and the shift factors to zero. Note that we use data augmentation and L2 weight decay of $10^{-5}$ to prevent overfitting instead of Dropout layers. In the training stage, we set both $\lambda_s$ and $\lambda_m$ to 0.001. In the fine-tuning stage, we aggregate losses at fidelity levels of 25, 50, 75, 100% to optimize multi-fidelity approximations.

**Channel Prioritization and Sparsity.** Figure 2 illustrates the scaling factors under different constraints. We use the Kendall's rank correlation Kendall (1955) to monitor if the scaling factors keep monotonicity along with the channel indices. Clearly, we can see monotonically decreasing initialization helps to maintain priority, sparsity regularization drives the scaling factors toward zeros, and monotonicity-induced penalty promotes monotonicity beside initialization. We visualize the compressed network structures in Figure 3.

**Multi-fidelity Approximations.** To test the computational efficiency, we compare with the pruning method proposed in Li et al. (2016) and Liu et al. (2017)[1]. It is shown that our method accomplishes better parameters reduction and FLOPs reduction rates. To test the adaptive computation ability, we evaluate the network at different fidelity levels and compare with another adaptive computation method, IDP, proposed by McDanel et al. (2017). On the CIFAR-10, our model at 25% fidelity level uses only 0.4% parameters and 3.2% FLOPs to achieve 75.9% accuracy, beating IDP by 65% accuracy. Note that IDP does not prune networks so that our approach also outperforms in both parameters and FLOPs reductions. The results are summarized in Table 1.

# 4 CONCLUSION

We propose a novel method for CNNs to learn prioritized channels, prune the network in a structure way, and incorporate multi-fidelity approximations to trade-off between varying resource and performance demands. Thanks to the cascading nature in deep neural networks, the input and output channels of an intermediate layer are both at $p \times 100\%$ fidelity level, so only $p^2 \times 100\%$ parameters are used in essence. Channel prioritization is a generalized training method that can be used to train or retrain any modern CNN that has BN layers, without modifying the network architecture or requiring extra hardware support.

---

[1]Implement the single-pass scheme and prune lower scaling factors by percentile threshold at 70%

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
