# OpenReview forum: "Channel-Prioritized Convolutional Neural Networks for Sparsity and Multi-fidelity"
_ICLR.cc/2018/Workshop — Reject_

### Official Review · AnonReviewer3 · 2018-02-27
**Interesting idea but baselines are very poor**

**Rating:** 4
**Confidence:** 4

**Review:**

The paper suggests an interesting approach to rank the importance of different channels by monotonically decreasing initialization of scaling factors in batch-norm layers. This potentially allows to order the channels wrt their importance and hence to establish a trade-off between evaluation time vs accuracy.

Although the idea is interesting I am not convinced it really works since all the VGG-16 baselines are very poor. On both CIFAR-10 and CIFAR-100 those architecture can be trained to much higher level of accuracy (93% for CIFAR-10 and about 79% for CIFAR-100). Because of this it is not clear whether further sparsification/switching to different fidelity levels will not lead to unacceptably bad quality.

---

### Official Review · AnonReviewer2 · 2018-03-04
**Interesting approach to obtain efficient inference based on pruning the batch norm**

**Rating:** 5
**Confidence:** 4

**Review:**

Pros:
-Somehow i like the idea of dynamically modify the computational cost depending on the inference resources available. This paper aims at that by using the so called 'fidelity'.

Cons:
- How do we select that fidelity? At random? And when it comes to multiple fidelities... how do we select that?
- The paper is introduced in a generic manner, however, this is based on a pre-trained effort.

Given said that, I do not see how this term is directly related to the computational resources available, and, it it is not, then probably it is better to just compress with different parameters. As far as I understand, everything needs retraining (and actually experiments are based on a pre-trained VGG16 and transferred to cifarX)

The idea of structured pruning using the batch norm layer is not strictly new and while seems to be benefits, I guess needs better explanations. For instance, the multi-fidelity seems interesting but is not really explained. How do I link inference benefits to having multiple fidelity?

I also wonder the impact in compression of the linear layers. Tables show the total number of parameters while figures are about convolutional layers. I understand compressing the last layer has a significant impact directly on the first linear layer, and therefore those numbers are a bit 'overselling'.

As minor comments, I also think the paper needs serious readability improvements. For instance, figures and tables are hard to read. Things are not defined properly. Took me a while to understand that, in Table 1, there is a direct comparison in each row between the proposed approach and the IDP (from the literature) and I still do not understand the differences between the rows when comes to 'ours'


In table 1, how can I compare the numbers for the model at 25% fidelity (Achieving 75.9 vs 65)? I can not see that in the table.

---

### Official Review · AnonReviewer1 · 2018-03-10
**simple effective approach for pruning cnns**

**Rating:** 7
**Confidence:** 3

**Review:**

This paper introduces a method of inducing sparsity over cnn channels by imposing sparsity loss and and a monotonicity inducing penalty on the batchnorm scaling factors for each layer. The end result is a network with many scaling factors close to zero, so these channels can be pruned and then the network is subsequently finetuned. Channel-wise sparsity is not a new approach, but the novelty in this method comes from the dynamics scaling idea. In particular in the finetuning phase they optimize for multiple fidelity level simultaneously, and then the model can be adapted to different resource requirements. The approach appears to outperform other recent methods of pruning cnns.

Specific questions:
- could you say something about other methods that prune chanel-wise? what would be the outcome of using another approach, and then pruning with different fidelity levels and then finetuning jointly for each level? more specifically I would like to get at the contributions of the particular sparsity loss used during training (sparsity + monotonicity), and how this relates to other chanel-wise sparsity penalties, and the multi-fidelity fine tuning.
- if you optimize for a single fidelity, can you achieve better performance (for that level) versus when you optimize for may levels simultaneously?
- could you say more about the relation between desenets (Huang et al., 2017)? this approach also learns networks of different capacities, albeit in a different manner by classifying an image earlier or later along a cascade of networks depending on the confidence. How does the test time efficacy of this approach compare to your method?

Overall this paper is clear, easy to read and presents clear improvements over existing pruning methods. Additional discussions to related models would be helpful to put this work in the context of previous pruning methods and make clear the specific contributions of this work.

---

### Decision · Program_Chairs · 2018-03-20
**ICLR 2018 Workshop Acceptance Decision**

**Decision:**

Reject

**Comment:**

Based on the reviews, this paper has not been accepted for presentation at the ICLR workshop. However, the conversation and updates can continue to appear here on OpenReview.